

# Mesospheric semidiurnal tides and near-12-hour waves through jointly analyzing five longitudinally-distributed specular meteor radars at boreal midlatitudes

Maosheng He[1] and Jorge L. Chau[1]

[1]Leibniz-Institute of Atmospheric Physics at the Rostock University, Kühlungsborn, Germany

**Correspondence:** Maosheng He (he@iap-kborn.de)

**Abstract.**

In the last decades, mesospheric tides have been intensively investigated with observations from both ground-based radars and satellites. Single-site radar observations provide continuous measurements at fixed locations without horizontal information whereas single spacecraft missions provide typically global coverage however with limited temporal coverage at a given location. In this work, combining eight years (2009-2016) of mesospheric winds collected by five specular meteor radars from three different longitudinal sectors at boreal midlatitudes ($49\pm8.5°$N), we develop an approach to investigate the most intense global-scale oscillation, namely, at the period $T=12\pm0.5$hr. Resolved are six waves: the semidiurnal westward-traveling tidal modes with zonal wavenumber 1, 2, and 3 (SW1, SW2, SW3), the lunar semidiurnal tide M2, and the upper and lower sidebands (USB and LSB) of the 16-day wave nonlinear modulation on SW2. The temporal variations of the waves are studied statistically with a special focus on their responses to sudden stratospheric warming events (SSWs), and on their climatological seasonal variations. In response to SSWs, USB, LSB, and M2 enhance, while SW2 decreases. However, SW1 and SW3 do not respond noticeably to SSWs, contrary to the broadly-reported enhancements in the literature. The USB, LSB, and SW2 responses could be explained in terms of energy exchange through the nonlinear modulation, while LSB and USB might previously have been misinterpreted as SW1 and SW3, respectively. Besides, we find that LSB and M2 enhancements depend on the SSW classification with respect to the associated split or displacement of the polar vortex. In the case of seasonal variations, our results are qualitatively consistent with previous studies, and show a moderate correlation with an empirical tidal model derived from satellite observations.

*Copyright statement.* TEXT

# 1 Introduction

The availability of observations limits the advance of studies on the mesosphere-lower-thermosphere (MLT). While in situ observations, e.g., with rockets, are available only on campaign bases, continuous observations can only be collected remotely




from either ground or space. Routine monitoring of the MLT is only possible with ground-based radars with all-weather applicability, and satellite-based optical instruments with good mobility.

Both ground- and space-based continuous observations have been used to investigate the global-scale MLT waves. Most of these studies were based on single-point analysis techniques and therefore were subject to inherent spatiotemporal ambiguities.

Ground-based observations from single radars could yield high-frequency-resolved spectra of MLT parameters, but can not resolve the global-scale structure (e.g., Azeem et al., 2000). On the other hand, space-based sensors, typically on-board slowly precessing polar satellites (e.g., Oberheide et al., 2002), collect data with global coverage but with limited temporal coverage for given locations. They are capable of determining the horizontal scales, which however cannot distinguish temporal variations from spatial variations. The obtained frequency spectra are usually Doppler shifted at limited resolution (e.g., Salby, 1982a, b).

To overcome the spatiotemporal ambiguity, specular meteor radars (SMRs) or medium frequency radars from multi-longitudinal sectors had been combined to resolve the horizontal scale of MLT waves at polar latitudes tentatively. A typical procedure is a least square regression (LSR) fitting of longitudinal harmonic functions with preassigned wavenumber to observations from different longitude sectors (e.g., Murphy, 2003). The LSR procedure was used to decompose the most significant global-scale periodicity, namely, the 12hr tidal oscillation, into the migrating mode SW2 (SW$m$ represents westward-traveling semidiur-

nal tidal modes with zonal wavenumber $m$) and nonmigrating modes SW1 and SW3 (mostly at polar latitudes, e.g., Murphy, 2003; Murphy et al., 2006, 2009; Baumgaertner et al., 2006; Manson et al., 2009). However, as sketched in Figure 1, such decomposition gets complicated by the existence of other waves in the vicinity of 12 hr with wavenumbers identical to those of solar tides. These include the semidiurnal lunar tide (M2), and the lower and upper sidebands (LSB and USB) of the nonlinear modulation of the 16-day planetary wave on SW2. Sharing similar periods and same wavenumbers with the tides, these waves

are suspected to have contaminated the interpretation of previous studies. Specifically, LSB and USB might have been detected at low-frequency resolutions and misinterpreted as SW1 and SW3 (cf., He et al., 2018a, b), respectively. Additionally, the M2 estimations might have been contaminated by LSB in spectral studies using single-site observational technique (as explained in He et al., 2017) or by the power leakage from SW2 in low-frequency-resolved spectral analyses (cf., Section 5.1 in He et al., 2018b).

The main purpose of the current study is to develop an approach to unambiguously separate all six waves sketched in Figure 1 using observations of five SMRs at latitudes near 49 °N between 2009 and 2016. Below, section 2 introduces the six waves and the approach. The results are shown in section 3, and used to investigate the six waves statistically, in particular, their responses to sudden stratospheric warming events (SSWs) and their seasonal variations (Sections 4 and 5, respectively). Note that in the current study, we use the term 'responses to SSWs' to refer to the behaviors associated with SSW, which does

not imply causative relations between the behaviors and the phenomenon suggested literally by the term 'SSWs', namely, the sudden increase in the temperature.





## 2   Data analysis

For the current study, we collect the mesospheric wind observations of SMRs at 49±8.5°N from three longitudinal sectors, namely, east Asian, Europe, and America. As shown in Figure 2, these SMRs are located at Juliusruh (12°E, 55°N, available since 2007), Collm (13°E, 51°N, since 2004), Beijing(116°E,40°N, since 2009), Mohe(123°E, 54°N, since 2012), and Tavi-

stock (81°W 43°N, since 2002). The radar system at Tavistock is officially known as Canadian Meteor Orbit Radar (CMOR, e.g., Jones et al., 2005). For details of the radars, e.g., working frequency, power, and configuration of antennas, readers are refer to Liu et al. (2016, 2017); Yu et al. (2013), Singer et al. (2013), Jacobi (2012), and Jones et al. (2005).

   The current study uses hourly zonal wind derived at a vertical resolution of 2 km according to the algorithm introduced by Hocking et al. (2001) and Stober et al. (2012). For each SMR, we filter oscillations in the wind at periods 11.6±0.1hr,

12.0±0.1hr, and 12.4±0.1hr, through high-frequency-resolved wavelet spectral analysis. For each period, we decompose the potential waves with different wavenumber by jointly analyzing the spectral coherency between the SMRs.

### 2.1   Decomposition approach

A $Morlet$ wavelet analysis is applied to the zonal wind at a given altitude for each SMR, resulting in spectra $\tilde{W}^n_{(f,t)}$ where $n=1,2,...,5$ represents the SMRs. $\tilde{W}^n_{(f,t)}$ corresponds to the phasor representation used (e.g., Murphy, 2002; Baumgaertner

et al., 2006). We attribute the coherence among $\tilde{W}^n_{(f,t)}$ to waves traveling in the longitudinal direction with zonal wavenumber $m_k$, ($k=1,2,..., K$) and complex amplitude $\tilde{a}_k$. At given $f$ and $t$, we fit $\tilde{\mathbf{a}}_k$ from $\tilde{W}^n_{(f,t)}$ following, e.g., equation 5 in He et al. (2018a),

$$\left(\tilde{W}^1, \tilde{W}^2, ..., \tilde{W}^5\right)' = \tilde{\mathbf{E}}_{5\times K}\tilde{\mathbf{a}}_{K\times 1} \tag{1}$$

Here, the $k$-th entry of $\tilde{\mathbf{a}}_k$ is defined as $\tilde{a}_k$, and the entry of $\tilde{\mathbf{E}}$ in the $k$-th row and $n$-th column is defined as $\tilde{\mathbf{E}}_{n,k} := e^{i2\pi m_k \lambda_n}$,

representing the phase of $k$-th wave detected by the $n$-th SMR at longitude $\lambda_n$. When $K<=5$, Equation 3 allows estimating $\tilde{\mathbf{a}}$ with preassigned $m_k$, as demonstrated in, e.g., Figure 4 in He et al. (2018a). Since our five SMRs are mainly from three distinct longitudinal sectors, our implementation entails $K \leq 3$. Although two of the five radars provide redundant information as they are in the same longitude sector as other SMRs, we use all five for a broader temporal coverage and higher statistical significance. We assign $m$ following Figure 1 for reasons detailed in Section 2.2.

Note that in estimating $\tilde{\mathbf{a}}$, we assume that the meridional variation of all waves is negligible among the SMRs. To test this assumption, we ran the the climatological tidal model of the thermosphere (CTMT, Oberheide et al., 2011) derived from TIDI and SABER . Semidiurnal components in the zonal wind at 50°N are highly correlated with those at 40°N: the correlation coefficients associated with SW2 and SW1 are 0.94 and 0.99, respectively (not shown here). For the latitude dependence of the semidiurnal tide and its seasonal variation, readers are referred to (e.g., Yu et al., 2015).

In principle, $\tilde{\mathbf{a}}$ could be estimated through the LSR or a short-time Fourier transform (STFT) within a sliding window (e.g., Murphy, 2003; Baumgaertner et al., 2006). Using a Gaussian window with the proper width, the LSR or STFT might even yield results identical to ours. The width of the window, $\Delta T$, determines proportionally the time resolution $\sigma_t \propto \Delta T$ , which



are coupled with the frequency resolution $\sigma_f$ according to the $Fabor$'s uncertainty principle $\sigma_t\sigma_f \geq \frac{1}{4\pi}$. The resolution in our wavelet analysis is determined by the $Morlet$ factor as specified in Section 2.2.

## 2.2 Targeting waves and assignment of zonal wavenumber

Tides are characterized by oscillations at periods which are integral fractions of a solar or lunar day. In the atmosphere, the solar
tides are primarily forced by daily variation in the absorption of sunlight (Chapman and Lindzen, 1970). At 12hr, the migrating component SW2 is known to be the dominant tide (e.g., Pancheva and Mukhtarov, 2012), while the nonmigrating components SW1 and SW3 are also frequently reported (e.g., Angelats I Coll and Forbes, 2002; Manson et al., 2009). At the latitude for our study (49°N), SW1 and SW3 are expected to be more intensive than other semidiurnal nonmigrating tides on climatological averages (not shown here) according to the tidal model (c.f., Oberheide et al., 2011). These solar tides, according to the classic
tidal theory (Chapman and Lindzen, 1970), have amplitudes ∼20 times larger than those of lunar-gravitationally-forced tides. Despite these theoretical predictions, oscillations at 12.4hr have been clearly detected in the upper atmosphere and ionosphere and explained as the lunar tide M2, particularly around SSWs (e.g., Stening, 2011; Fejer et al., 2011; Chau et al., 2015). The occurrence of M2 was also confirmed by a wavenumber identification using a dual-SMR network ($m$=2 at 12.4hr during SSW 2013, cf., He et al., 2018b). The significant M2 tide was attributed to the lunar forcing resonance due to a shift of a local
maximum (namely the so-called $Pekeris$ peak) in the atmospheric frequency response, which is supported by a comparison in a numerical experiment using GSWM driven by two specifications of a climatological-mean background atmosphere and that during SSWs (Forbes and Zhang, 2012).

In addition to the M2, also oscillating at the period 12.4 hr is a westward-traveling structure with zonal wavenumber $m$=1, namely, the lower sideband (LSB) of the modulation of the 16-day planetary wave (PW) on SW2 tide (as explicitly detected
and explained in He et al., 2018a). LSB's $m$ and $f$ are determined by their parent waves according to the nonlinear interaction resonance conditions $\tilde{\Psi}_{LSB} = \tilde{\Psi}_{SW2}\tilde{\Psi}^*_{PW}$. Here, $\tilde{\Psi}_{\bullet} := e^{i2\pi(f_{\bullet}t+m_{\bullet}\lambda)}$ represents the phase of a wave ● (e.g., He et al., 2017). The 16-day PW is a normal wave, and its intrinsic period of 12.5d is determined by the resonant properties of the atmosphere (e.g., Ahlquist, 1982; Longuet-Higgins, 1968; Madden, 2007; Salby, 1984). Doppler-shifted by the prevailing eastward wind during winter, the PW is observed at a period up to 20 days, with an average of 16 days (for the climatology of the 16-day
PW, cf., Luo et al., 2002; Day and Mitchell, 2010). The corresponding LSB occurs in the frequency range of $f^{LSB}$= (2-1/12.5, 2-1/20) $d^-$, associated with LSB at $T^{LSB}$=12.4±0.1hr. Similar to LSB, an upper sideband (USB), at $T^{USB}$=11.6±0.1hr and $m$=3, might also be excited by the modulation, following the resonance conditions $\tilde{\Psi}_{USB} = \tilde{\Psi}_{SW2}\tilde{\Psi}_{PW}$ (as explicitly detected in He et al., 2018a). To encompass the most potential LSB and USB periods, in our wavelet analysis (cf., Grossmann et al., 1990), we set the $Morlet$ factor to 128 so that the passed frequency band corresponds to 12.4±0.1hr, 11.6±0.1hr, and
12.0±0.1hr.These period bands are narrow enough to prevent power leakage between each other.

In Figure 1, we assume that at 12.0hr the most important waves are tides SW1, SW2 and SW3 ($m_1$=1, $m_2$=2 and $m_3$=3), at 12.4hr M2 and LSB are dominant ($m_1$=1 and $m_2$=2), and at 11.6hr exist only the USB ($m_1$=3). With these assignments of $m$ and according to Equation 3, we repeat the estimation of $\tilde{a}$ on the grids of date $t$ and altitude $h$ at each of the three





periods, resulting in the amplitudes for all six waves, $\tilde{a}^{\bullet}(h,t)$, where $\bullet$ represents LSB, M2, the USB, SW1, SW2, or SW3. The corresponding amplitude $|\tilde{a}^{\bullet}(h,t)|$ is displayed in Figure 3.

## 3  Results

In Figure 3, the decomposition is based on observations from five SMRs between 2012 and 2016, whereas before 2012 only four SMRs are available (Mohe SMR started operation in 2012). The different SMR combinations are designated by the yellow and cyan lines at the bottom of Figure 3f. Using the four SMRs, we also produced the results between 2012 and 2016, which are highly consistent with the results from the five SMRs: the corresponding correlation coefficients are 0.92, 0.96, 0.93, 0.95, 0.98, and 0.95 for the six components, respectively. In Figures 3a-c, the horizontal yellow line around January 2013 shows that the amplitudes are quantitatively consistent with the recent estimation using only the two SMRs at Juliusruh and Mohe: the components $m=1$, 2, and 3 maximize at roughly 4, 8, and 8m/s in both Figures 3a-c here and Figure 4 in He et al. (2018b). The correlation and consistency suggest that the decomposition is overly not sensitive to the absence of one SMR between 2009 and 2011.

The temporal variations in Figures 3a-f share some similarities. First, in Figures 3a-c LSB, USB, and M2 are often enhanced noticeably in the month following the vertical magenta dashed lines which indicate the polar vortex weakening (PVW, cf., Zhang and Forbes, 2014) as a reference of SSWs in the current study. Second, as shown in Figures 3d-f, SW1, SW2, and SW3 are characterized by repeating annual patterns, as separated by the calendar new year indicated by the solid white lines. For a statistical study on the SSW responses and the seasonal variations, we average the amplitudes of the six components with respect to the time since the PVW epoch and the calendar new year, following the composite analysis approach (CA, e.g., Chau et al., 2015). CA is also known as superposed epoch analysis, SEA, in geophysics and solar physics (e.g., Chree, 1914). The PVW and calendar results are shown in Figure 4 and discussed in Sections 4 and  5, respectively.

## 4  Responses to SSWs

As the most radical manifestation of stratosphere-troposphere coupling, SSWs impact the upper atmosphere in broad altitude and latitude ranges (e.g.,  Goncharenko and Zhang, 2008; Goncharenko et al., 2013). One type of impact is the broadly-reported enhancements of various waves at periods near 12 hr, including M2, SW1, SW3 and LSB and USB (e.g.,  Chau et al., 2015; Angelats I Coll and Forbes, 2002; Liu et al., 2010, and references therein). Recently, He et al. (2017) argued that there might not be SW1 and SW3 enhancements during SSWs and instead suggested that the reported enhancements are just misinterpreted signatures of LSB and USB at low-frequency resolution, respectively. These arguments about SW1 and SW3 were supported observationally by two case studies (He et al., 2018a, b, , respectively). Here, we extend this earlier interpretation statistically in Section 4.1 and investigate their year-to-year variability in Section  4.2.



### 4.1 Multi-year average

The SSW CA results in Figures 4a-f suggest that among the six components, only three, namely, LSB, M2 and USB, exhibit a sharp maximum immediately following PVW, whereas the others, namely, SW1, SW2 and SW3, do not: their intensities are largely decreasing from 40 days before PVW to 50 days after. The enhancements of LSB, USB, and M2 around SSWs are consistent with existing studies, both statistical studies with single radar approaches (e.g., Chau et al., 2015) and case studies (e.g., He et al., 2017). However, our finding that SW1 and SW3 do not show enhanced intensity during SSWs are at variance with most existing studies(e.g., Liu et al., 2010; Pedatella and Forbes, 2010; Pedatella et al., 2012; Pedatella and Liu, 2013; Wu and Nozawa, 2015). LSB and USB enhancements associated with non-enhancing SW1 and SW3 support the hypothesis that LSB and USB were detected at low frequency-resolution and misinterpreted as SW1 and SW3, respectively (He et al., 2017). In a case study on SSW 2009, evidence for the SW1 misinterpretation was extracted with an intercontinental-scale dual-SMR network extending along 80°N(He et al., 2018a), while in another case study on SSW 2013, similar evidence was identified for the SW3 misinterpretation with a similar network at 54°N(He et al., 2018b). Here, we report the first multi-year statistical evidence. Besides the responses of LSB and USB, also supporting the hypothesis is the decreasing SW2 decrease at PVW (note that the color is scaled for SW2 amplitude in a range broader than those of others). The declining SW2 feeds the LSB and USB enhancements: SW2 provides 100% and 97% of the energy of the LSB and USB, respectively, according to the $Manley - Rowe$ relations detailed in He et al. (2017).

### 4.2 Year-to-year variability during SSW

Although LSB, USB, and M2 composite behaviors look similar to each other in Figures 4a-c, their patterns show remarkably different year-to-year variability as shown in Figures 3a-c. To investigate the year-to-year variability, we conduct a CA similar to $\langle|\tilde{a}|\rangle$ displayed in Figures 4a-f, but for the complex amplitude $\langle\tilde{a}\rangle$. Different from the $\langle|\tilde{a}|\rangle$ in Figures 4a-c where all three components maximize during SSW, in $|\langle\tilde{a}\rangle|$ (not shown here) only M2 maximizes whereas LSB and USB do not. Determined by the phases of both SW2 and the PW at SSWs, $\langle\tilde{a}\rangle$ of LSB and USB exhibit more randomness than that of M2 whose phase is determined only by the M2 phase at SSW. The consistence between $|\langle\tilde{a}\rangle|$ and $\langle|\tilde{a}|\rangle$ of M2 might be attributed either to a potential association between SSW and a particular lunar phase (as suggested by, e.g., Fejer et al., 2010) or simply to the limited sampling number of M2 enhancement events during SSWs (see Figure 3b).

To explore possible relationships between the enhancements of different waves, we search, in Figures 3a-c, the maximum amplitude in a 30d-wide window following each PVW, as a measure of the intensity of the corresponding enhancements. The maxima are marked as magenta plus symbols in Figure 3. A clear association is found between LSB and M2 enhancements. As illustrated in Figure 5, the seven events are clustered mainly into three groups. In the case of other combinations, i.e., USB vs. M2, or LSB vs. USB, we have not found any noticeable relationship. This result suggests that LSB and USB are independent of each other during SSWs. The lack of coupling between the sidebands has been discussed in detail in Section 4.5 in He et al. (2017).



We further investigate three clusters in Figure 5 according to a classification of associated SSWs (cf., Seviour et al., 2016; Esler and Matthewman, 2011): vortex-split or displacement marked by black solid and open circles, respectively. Clearly, three clusters circled in the magenta lines in Figure 5 are associated with the SSW classification: (a) the strongest LSB and intermediate M2 occur in vortex-displacement events, (b) intermediate LSB and strongest M2 occur in vortex-split events, and

(c) weakest M2 and weakest LSB occur mostly in non-SSW events. The only exception in this classification is the 2015 event. The association between the SSW classification and M2 strength is consistent with the conclusion drawn from more SSW events using equatorial magnetic field observations (e.g., Siddiqui et al., 2018). Here, our multi-SMR-jointed analysis allows us to separate LSB and M2 components that share the same period. Our results imply that LSB has contaminated previous M2 estimations based on single-site observations, particularly during vortex-displacement SSWs.

**5   Climatological seasonal variations of the solar tides**

In the current section, we change our focus to the seasonal climatology of the identified six waves. Similar to Figures 4a-f showing the SSW CA with respect to PVW, Figures 4g-l display the CA results with respect to the start of the calendar year. Figures 4g-l exhibit similarities with Figures 4a-f, e.g., similar vertical and temporal extensions of the primary peaks. The similarities are not surprising since the time epochs are close to each other: PVWs always occurred in winter near the

start of the new year. In comparison with the SSW CA results, in the calendar CA the primary peaks of LSB, USB, and M2 (Figures 4g-i) are slightly smeared out. In contrast, the peaks of the solar tides (SW1, SW2, and SW3 in Figures 4j, 4k, and 4l, respectively) have not been smeared out in the calendar CA, the peaks of SW2 and SW3 are even sharper and stronger. These results suggest that the temporal variations of LSB, USB, and M2 dominated more by their responses to SSWs than by their seasonal variations, whereas those of the solar tides are characterized more by the seasonal variations.

**5.1   Comparison to previous studies**

In the amplitude plots shown in Figures 4j-l and 3d-f the vertical variations are characterized by larger amplitudes at higher altitudes. MLT waves are often excited in and propagated from the stratosphere or troposphere. The upward propagating waves amplify exponentially with increasing altitude as the air density decreases. It follows from such a simple vertical structure that the temporal variations are similar at all altitudes, allowing the two-dimensional variations in Figures 3d-f to be described

largely as one-dimensional temporal variations. Accordingly, we average vertically the amplitudes shown in Figures 3d-f, and display the average as a function of the day of year (DoY) in Figure 6a. The averaged components are shown as scatter plot in Figures 6b, 6c, and 6d, against each other, i.e., for SW1 vs SW2, SW3 vs SW2, and SW3 vs SW1, respectively. The most salient feature in Figure 6 is that SW2 is almost always the dominant component, except in late October when SW3 is comparable to SW2. These comparisons suggest that the 12.0hr harmonic amplitude (S2) from single radar analyses is overall

a reasonable approximation of SW2 (as used in, e.g., Conte et al., 2018) except between October and November.

In Figure 6a, the red, green and blue solid lines display the vertical average of the CA results shown in Figures 4j-l, which summarize the main seasonal variations of the solar semidiurnal tides: SW1, SW2, and SW3, respectively. SW2 is characterized



by two comparable peaks in September and in December, and steep decreases in September-October and March-April (DoY 250-300 and 0-80), consistent with Conte et al. (2018). SW1 is characterized by a single peak appearing in winter and a minimum in summer, which is largely consistent with thermospheric seasonal variation of SW1 at 50°N according to CHAMP observations (Figure 12, in Oberheide et al., 2011). SW3 is characterized by two peaks in earlier May and October (at DoY 130 and 280), respectively. Similar annual dual peaks of SW3 were observed from SABER measurements (Figure 2.7 in Hartwell, 1994), and also obtained at 50°N at 88km altitude from the three years (2006-2008) of simulated data from the Canadian Middle Atmosphere Model Data Assimilation System (Figures 6b and 10c in Xu et al., 2012). Interestingly, in Figure 6a, d, the relative importance of SW1 and SW3 switches around early April and November (DoY 90 and 310) : in summer SW3 is stronger than SW1 but SW1 is stronger in winter. These seasonal variations might be associated with the climatologies of the background mean wind (e.g., Laskar et al., 2016; Conte et al., 2017).

In comparison with some previous studies, the amplitudes in Figure 3 appear to be weaker (e.g., Jacobi, 2012), for at least three potential reasons. First, based on single-site observations, most existing studies did not separate waves with different wavenumbers but have to explain the total oscillations at 12hr or 12.4hr as approximations of SW2 or M2 (e.g., Chau et al., 2015). Second, most existing studies used windowing functions much narrower than ours, resulting in broader passbands and capturing more energy (e.g., Chau et al., 2015; Forbes and Zhang, 2012). Third, some studies present the amplitude of total wind including both zonal and meridional (e.g., Chau et al., 2015; Conte et al., 2017) while here we focus only on the zonal component. For a quantitative comparison, in the next section, we present a comparison with an independent empirical tidal model CTMT.

## 5.2 A comparison with an empirical model

Figures 7a-f present composite analysis in the same manner as in Figures 4j-l but for the amplitude of complex average, $|\langle \tilde{a} \rangle|$. The similarities between Figures 4j-l and 7a-c, indicate that the phases of solar tides are consistent from year to year.

For an independent quantitative comparison, we present the seasonal variation of the solar tides according to CTMT (Oberheide et al., 2011), in Figures 7g-l. The model results exhibit some consistency with our results, especially on SW2. SW2 in Figure 7h maximizes during August-September and December-January, in between these periods there is a minimum. The vertical gradient is steeper during the August-September maximum than that during December-January These features are similar to that in Figure 7b. However, in the model results the maximums or the minimum can hardly be observed (Figure 7h)).

These discrepancies might arise from the low temporal resolution of the model. The effective resolution is about two months during which the satellite observations used for the model cover the whole local time once. In contrast, the September maximum and the minimum are narrower than two months, therefore they might be smeared out. In the case of phase, our result in Figure 7e also exhibit similarities as the model results in Figure 7k.

SW3 is compared in Figures 7c, 7f, 7i and 7l, from which similarities in both amplitude and phase occur mainly in fall. Although SW3 in the model results also exhibit a second maximum, it occurs during February-March, up to two months different from the second annual peak in early May from our results (see Figures 7c and 6a). This difference might be associated



with the seasonally uneven sampling of observations used for the model: the satellite takes two months to cover all local time sectors. In the case of SW1, major discrepancies are found in both amplitudes and phases.

These qualitative findings are supported quantitatively by Figures 8a, 8b, and 8c where the in-phase and quadrature components of our estimations versus model are shown as scatter plots for SW1, SW2, and SW3, respectively. The highest correlation is observed in SW2. In the next section, we check if the discrepancies between our estimation and the model are caused by the failure of our main assumption used in our approach introduced in Section 2.

### 5.3 Bias of our estimation due to the existence of neglected solar tidal components

For estimating the solar tides and as explained in Section 2.2, in our approach we assume that our targeting components, i.e., SW1, SW2, and SW3, are the dominantly components at 12.0hr. This assumption might be too strong given that other neglected semidiurnal tidal components have been also reported (e.g., Oberheide et al., 2011; He et al., 2011). The current section quantifies the bias due to the existence of other neglected solar semidiurnal tidal components according to CTMT.

Arrange Equation 3 into two parts, namely, the targeting components with amplitudes $\tilde{\mathbf{a}}^{tar}_{3\times1}$ and the neglected components with $\tilde{\mathbf{a}}^{neg}_{(K-3)\times1}$,

$$\left(\tilde{W}^1, \tilde{W}^2, ..., \tilde{W}^5\right)' = \tilde{\mathbf{E}}_{5\times K}\tilde{\mathbf{a}}_{K\times1} := \tilde{\mathbf{E}}^{tar}_{5\times3}\tilde{\mathbf{a}}^{tar}_{3\times1} + \tilde{\mathbf{E}}^{neg}_{5\times(K-3)}\tilde{\mathbf{a}}^{neg}_{(K-3)\times1} \tag{2}$$

Multiply $(\tilde{\mathbf{E}}^{tar}_{5\times3})^{-1} := \left((\tilde{\mathbf{E}}^{tar}_{5\times3})^T \tilde{\mathbf{E}}^{tar}_{5\times3}\right)^{-1}(\tilde{\mathbf{E}}^{tar}_{5\times3})^T$, resulting in,

$$(\tilde{\mathbf{E}}^{tar}_{5\times3})^{-1}\left(\tilde{W}^1, \tilde{W}^2, ..., \tilde{W}^5\right)' = \tilde{\mathbf{a}}^{tar}_{3\times1} + (\tilde{\mathbf{E}}^{tar}_{5\times3})^{-1}\tilde{\mathbf{E}}^{neg}_{5\times(K-3)}\tilde{\mathbf{a}}^{neg}_{(K-3)\times1} \tag{3}$$

Here, the term on the left side is the estimated amplitude of the targeting components, while the first term on the right is the corresponding real amplitude. Therefore, their difference, i.e., the second term on the right, corresponds to the bias due to the neglected . According to CTMT, we estimate the bias and display its absolute value in Figure 9. Contributing to the bias are semidiurnal components SE3, SE2, SE1, S0, and SW4. The bias is overall below 2 m/s, except above 90km in summer, which suggests our main conclusions in previous sections are not affected by our assumption that SW1, SW2 and SW3 are the dominant components. Actually, determined by the configuration of the SMRs, the matrices of both $(\tilde{\mathbf{E}}^{tar}_{5\times3})^{-1}$ and $(\tilde{\mathbf{E}}^{tar}_{5\times3})^{-1}\tilde{\mathbf{E}}^{neg}_{5\times(K-3)}$ are very well-conditioned (with condition number of 1.6 and 2.4, respectively), suggesting our estimations are not sensitively affected by the errors in both the wavelet spectra and the neglected semidiurnal components.

Our comparison from the previous section has stressed the additional information that our results bring, specifically, those on SW1 and SW3. In future efforts, we plan to add more ground-based observations, and try to combine them with satellite-based wind and temperature observations (cf, Zhou et al., 2018), to improve our understanding of mesospheric tides. Although the current work focuses on the near 12 hr waves at midlatitudes, our joint-dataset analysis approach could be extended to other periods, e.g., diurnal or terdiurnal tides.

## 6 Conclusions

Combining mesospheric zonal wind observations from five midlatitude SMRs, we develop an approach to investigate statistically six waves at periods close to 12 hr, namely, three solar tides (SW1, SW2, and SW3), two sidebands of nonlinear modulation of 16-day wave on SW2 (LSB and USB), and a lunar tide (M2). We first filter the observation from each SMR into three narrow frequency bands through a high-frequency-resolved wavelet analysis. Then, in each of the three bands, wavelet spectra from five SMRs are combined to fit the potential waves. The results suggest that the temporal variations of the waves are characterized by responses to SSWs (enhancements of LSB, USB, and M2, and a decrease in SW2) and climatological seasonal variations of the solar tides. Our main results are:

(1) Contrary to extensive previous literature, SW1 and SW3 do not enhance during SSWs. Our results suggest that the LSB and USB enhancements have been misinterpreted as SW1 and SW3 signatures, respectively. Meanwhile, the enhancements are associated with a decrease in SW2, which could be explained in terms of the energy exchange through the nonlinear interaction.

(2) Both enhancements of LSB and M2 depend on the SSW classification with respect to the polar vortex split or displacement. M2 enhancement is stronger during vortex-split SSWs than that during the vortex-displacement, whereas LSB is the other way around. Overall, M2 is stronger than LSB, except during the vortex-displacement SSW when they are comparable, implicating that LSB might contaminate the existing M2 estimations based on single-site observations.

(3) The seasonal variations of solar tides are in reasonable agreements with existing observational studies: SW2 is the dominant component which maximizes around September and December followed by two minimums; SW1 maximizes in winter; and SW3 maximizes in fall and spring. In October when SW3 is at its annual maximum and SW2 is at a minimum, when their strengths are comparable to each other. These results suggest that the 12.0hr harmonic amplitude from single radar analyses is a reasonable approximation of SW2 for most of the time but not always.

*Code availability.* TEXT

*Data availability.* TEXT

Our main results, namely, the complex amplitudes of the six waves as a function of time and altitude, are shared at ftp://ftp.iap-kborn.de/data-in-publications/HeACP2018/.

*Code and data availability.* TEXT

*Sample availability.* TEXT





*Author contributions.* TEXT

Conceptualization, M.H. and J.L.C.; Methodology, M.H.; Software, M.H.; Writing– original draft, M.H.; Writing– review&editing, M.H. and J.L.C.; Funding acquisition, J.L.C..

*Competing interests.* TEXT

5      The authors declare having no conflict of interest.

*Disclaimer.* TEXT

*Acknowledgements.* The authors appreciate the suggestions from Weixing Wan of jointly analyzing SMR observations from different longitudinal sectors, the discussions with Peter Hoffman on the tidal climatologies, and the discussions with Nick Pedatella and Jens Oberheider on the error estimation. We thank Guozhu Li for operating the SMRs at Mohe and Beijing, Christoph Jacobi for the data from Collm SMR, and Peter Brown for the CMOR SMR data. We are also grateful to Gunter Stober for processing the hourly wind from SMRs. This study is partially supported by the WATILA project (SAW-2-15-IAP-5 383) and by the Deutsche Forschungsgemeinschaft (DFG, German Research Foundation) under SPP 1788 (DynamicEarth) project CH 1482/1-1 (DYNAMITE). The SMR data from Mohe and Bejing are provided by BNOSE (Beijing National Observatory of Space Environment), IGGCAS (Institute of Geology and Geophysics, Chinese Academy of Sciences) through the Data Center for Geophysics, National Earth System Science Data Sharing Infrastructure (http://wdc.geophys.cn). The model CTMT (Climatological Tidal Model of the Thermosphere) is available at http://globaldynamics.sites.clemson.edu/articles/ctmt.html.



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





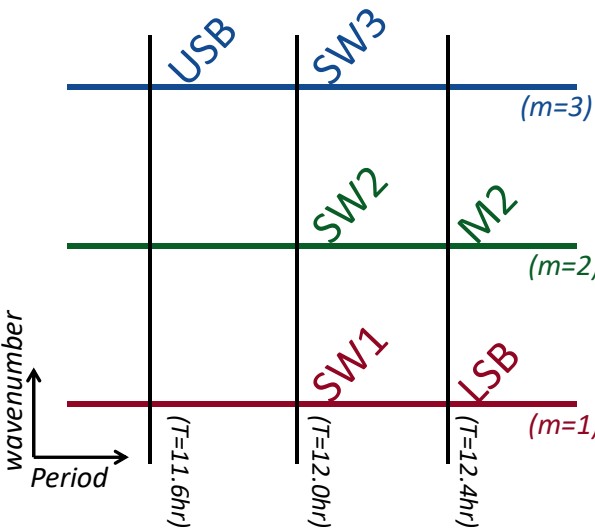

**Figure 1.** Distribution of near-12hr waves in the frequency and zonal wavenumber space (adapted from He et al., 2018a). In the current study, the colors red, green, and blue represent waves with zonal wavenumber $m$=1, 2, and 3, respectively.

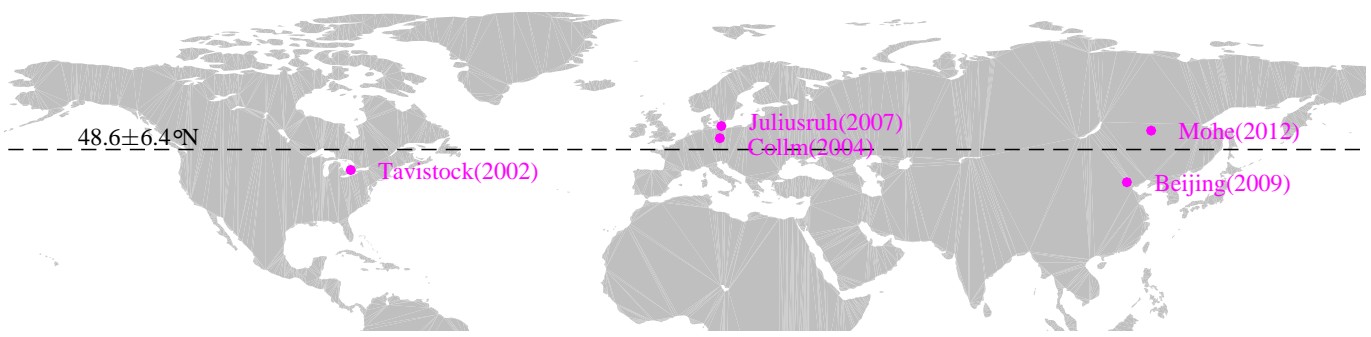

**Figure 2.** Distribution of five SMRs used in the current study. The numbers following the location names present the earliest available years of the corresponding observations.





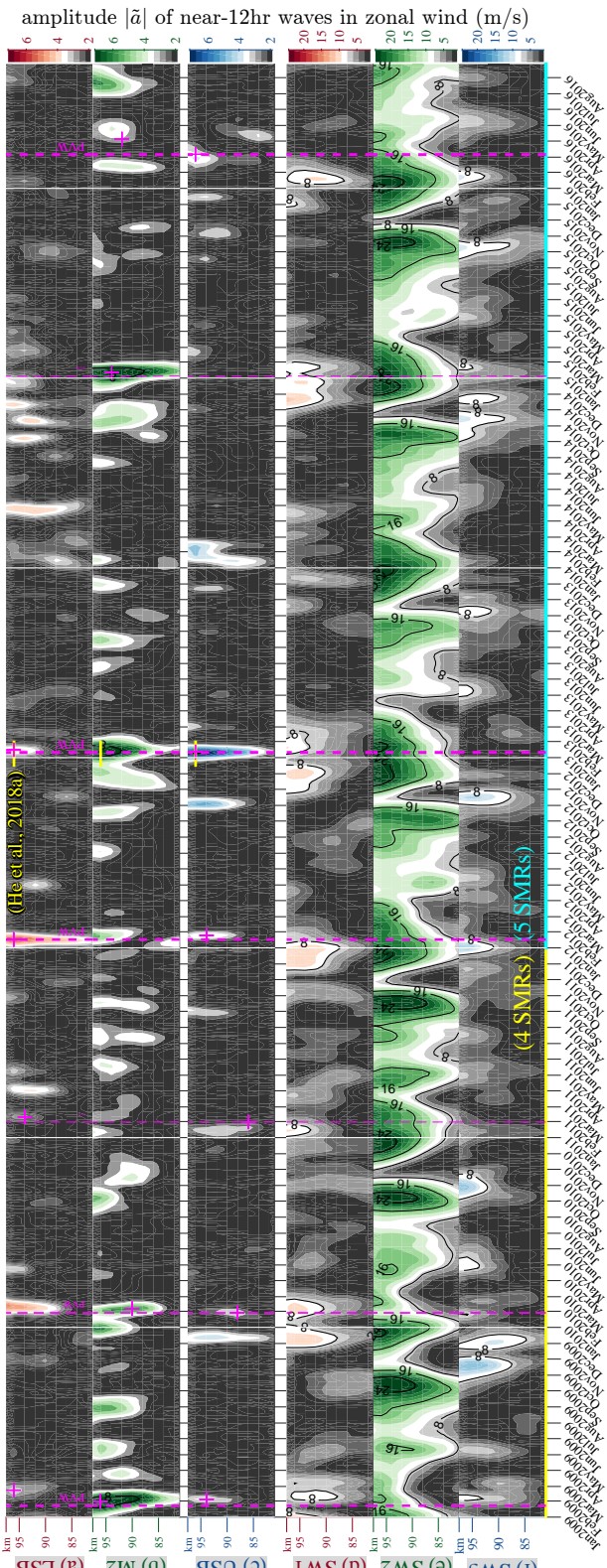

**Figure 3.** (a) The amplitude of the lower sideband (LSB) of the nonlinear modulation of the 16-day wave on the semidiurnal tide SW2 as a function of time and altitude. (b-f) The same plots as (a) but for the lunar tide M2, the upper sideband (USB), and the solar tides SW1, SW2, and SW3, respectively. (g) The altitude averages of Panels a, b, and c (LSB, M2, and USB), and (h) those of Panels a, b, and c (SW1, SW2, and SW3). In each Panel, the solid white vertical lines display the first day of each year, and the dashed magenta lines display PVWs. In (f), the cyan line on the bottom illustrates the interval from 2012 to 2016 in which all decomposition is based on five SMRs whereas the yellow line represents that MSR observations are not available at Mohe. In (a-c), the magenta plus symbols illustrate the maximum amplitude in each 30-day window following each PVW.





**Figure 4.** (a) Composite analysis of LSB from Figure 3a with respect to the occurrence of PVWs, namely, the magenta dashed lines in Figure 3a. (b-f) Same plots as (a) but for M2, USB, SW1, SW2, and SW3 from Figures 3b-f, respectively. (g-l) Same plots as (a-f) but with respect to the start of the calendar years, namely, the white lines in Figure 3.





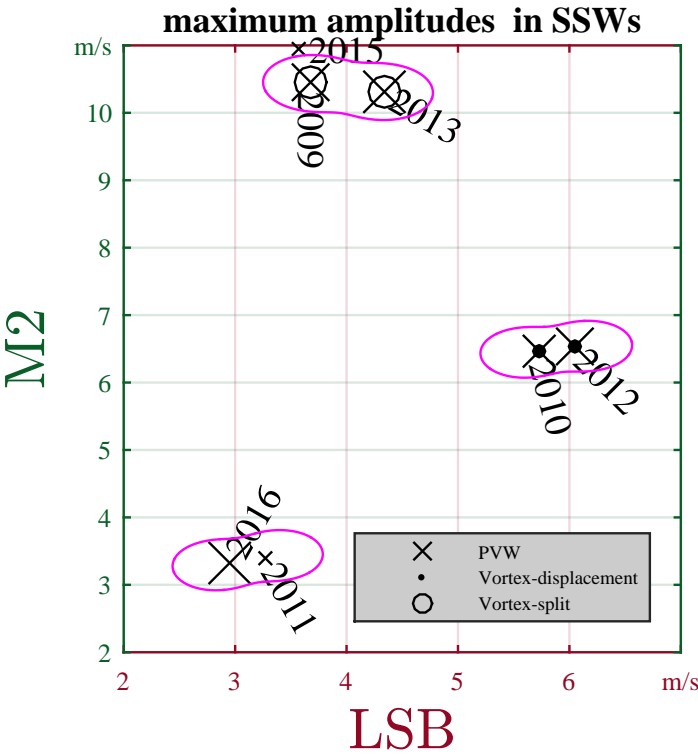

**Figure 5.** Scatter plots of the maximum amplitudes of LSB and M2 during SSWs, read from Figure 3. The size of the cross is proportional to the PVW strength. The magenta circles cluster the PVWs into three main groups according to the SSW classification according to the associated polar vortex split or displacement.



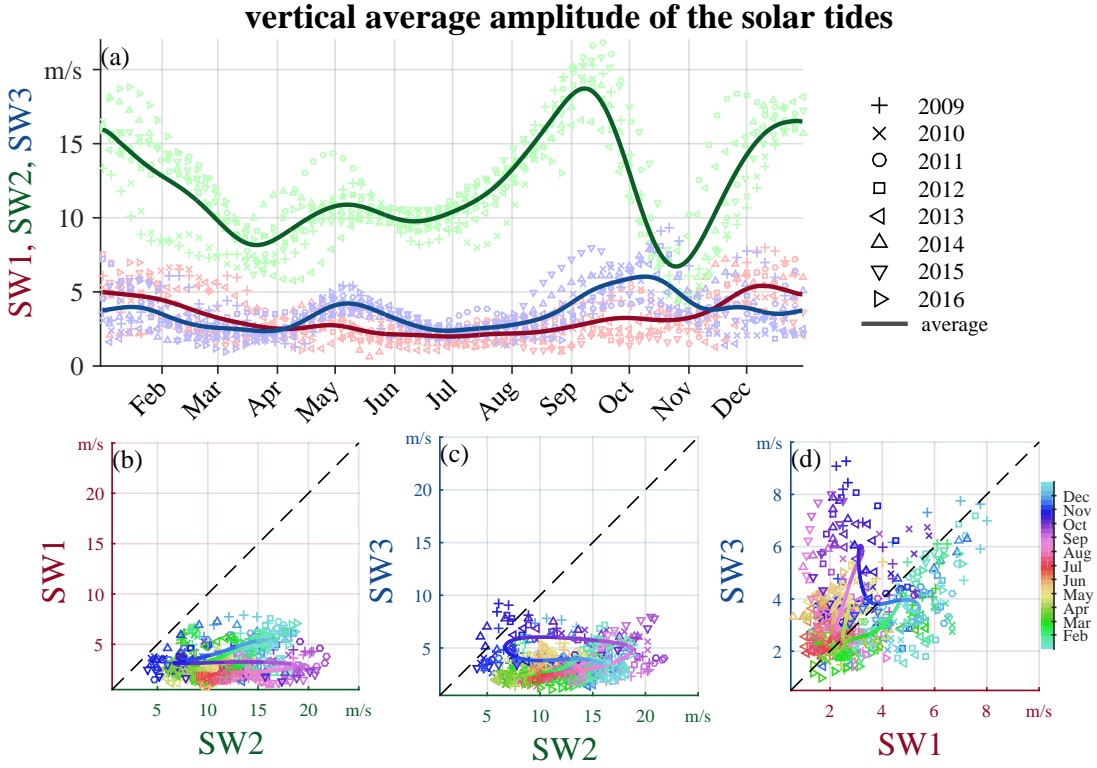

**Figure 6.** (a) Vertical average amplitude of SW1, SW2, and SW3 scattered as a function of date. (b, c, d) Scatter plot between the vertical average amplitudes of SW2 vs SW1, SW2 vs SW3, and SW1 vs SW3, respectively. In each panel, each point corresponds to a five day interval in Figure 3; and the solid colored line represents the multi-year average.





**Figure 7.** (a, b, c) Same plots as Figures 4j, 4k, 4l but for complex amplitude $|\langle\tilde{a}\rangle|$ of SW1, SW2, and SW3, with their phase shown in (d, e, f), respectively. (g-l) Similar plots as (a-f) but according to the climatological tidal model of the thermosphere (CTMT) derived from SABER and TIDI observations (Oberheide et al., 2011).



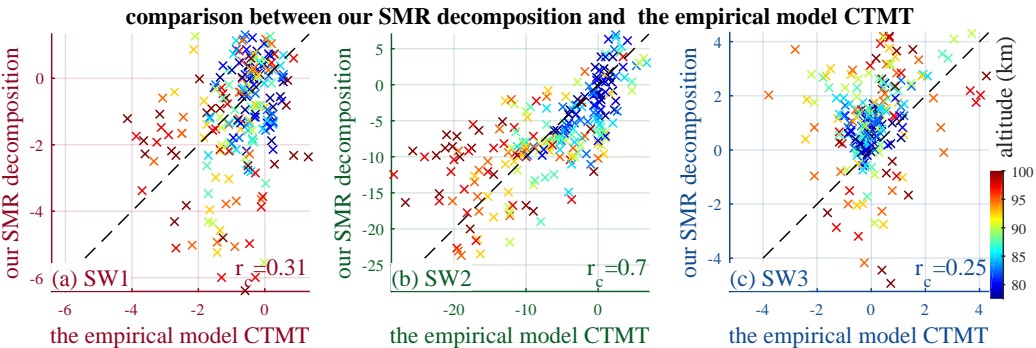

**Figure 8.** Scatter plots of SW1,SW2 and SW3 shown in Figures 4a-f, against those shown in Figures 4g-l. Each cross in Panel (a,b,c) represents the real or imaginary part of one pixel in Figures 7g, 7h, 7i, respectively.

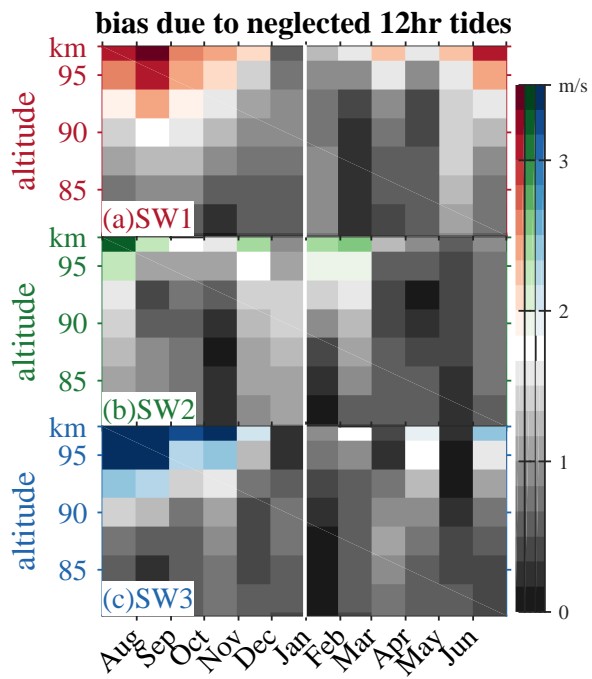

**Figure 9.** The bias of our tidal estimation due to the existence of neglected 12hr tides (including SE3, SE2, SE1, S0, and SW4) predicted by CTMT (Oberheide et al., 2011).