# Peer review of "Mesospheric semidiurnal tides and near-12-hour waves through jointly analyzing observations of five specular meteor radars from"

_Atmospheric Chemistry and Physics, 2018_

## Referee Comment (RC1) · Anonymous Referee #1 · 8 Mar 2019

Review of *Mesospheric semidiurnal tides and near-12-hour waves through jointly analyzing five longitudinally-distributed specular meteor radars at boreal midlatitudes* by He and Chang.

This manuscript reports on 6 semidiurnal and cuasi-semidiurnal tidal components, namely, SW2, SW1, SW3, M2 and the lower and upper side band interactions of SW2 and the 16-day PW (LSB and USB). These are derived from 5 radar measurements located at roughly three longitudes and ~50N. The three 12hr components are further compared to results from CTMT, showing a relatively good agreement. They also study the tidal enhancement during SSWs. As in previous works, the authors suggest that, due to their close period, aliasing between these waves may have lead to misinterpretation in semidiurnal tide measurements.

The paper is well written and easy to read, the methodology is generally adequate and the results are convincing. In general, the conclusions are not completely new and the results confirm previous works, but, in this one, a decomposition of the six waves is simultaneosly performed using several radar measurements extending 7 years.

I think the manuscript deserves publication in ACP once the following comments are addressed.

Main comments

1. The title is misleading. It is true that 5 radars are used but they are not longitudinally distributed: only 3 longitudes are sampled. Please, change the title accordingly.

2. Radar measurents have a significantly better resolution than CTMT results (~2months). Comparisons at similar temporal resolutions would make more sense. In that sense, the discussion in the text and panels a-f in Fig. 7 accordingly. On the other hand, that would additionally allow to delete Fig. 7 a-c panels, which are almost the same as j-l in Fig.4. That the phase does not change from year to year could just be mentioned in the text.

3. The authors report here (and also in previous works) that SW1 and SW3 are aliased with LSB and USB, respectively, in most previous tide studies from observations. I guess that CTMT is also that case. In the radar measurements with CTMT comparisons, why do you then compare SW1 and SW3 for both datasets instead of SW1+LSB and SW3+USB for the measurements with SW1 and SW3, respectively, for the model?

4. In general, the manuscript sometimes misses the opportunity to explain reasons for the tidal behavior. For example, there is no mention of the origin of the tidal seasonal variation. Also, the reasons for the LSB and M2 dependence on the SSW classification could be explored further. Potential attribution to planetary wave of particular wavenumbers might help.

Specific comments

P2L1-2. MLT monitoring is not only possible with those two techniques. Please, expand.

P2L4. Please, write single-point (in space or time).

P8L8-9. this is not exact. Slowly precessing satellite measurements may have large temporal resolutions but still can distinguish temporal variations. Also, finer temporal resolution can be achieved in some particular cases (e.g. Li et al., JGR, doi:10.1002/2015JA021577, 2015).

P3L2. Are the data available continuosly from the years indicated?

P3L16. Define f and t

P3L28. Even if the correlation between 40 and 50N is high, what is the difference in amplitude of these modes in CTMT?

P4L1. are -> is

P3L5. 12hr -> 12hr period,

P4L31.12.0h period

P4L32. Shortly explain in the text why only those m_k 's

P5L2. Please, comment on possible aliasing with other period waves.

P6L28. Is it possible that the interaction between PW and M2 is the origin of LSB?

P6 Why didn't you consider minor wamings?

P7L3. Please, provide your definition of PVW strength.

P7L5. Is there any relationship between PW1 (associated to displacement events) and the strong LSB (m=1)?

P7L5. There was a major final warming in March 2016 (Manney et al., 2016) but you find weakest M2 and LSB at that time. Please, explain why. Perhaps indicating your definition of non-SSW would help.

P7L18. I do not think it suggest to be more dominated by SSW but just that SSW have a significant effect.

P7L23. Please, specify that waves eventially dissipate.

P7L24. temporal variations are similar except when the altitude of dissipation changes with season. In general, it seems that does not apply to your waves (except SW3 in December, which appartently starts dissipating at lower altitudes than other years. Please, comment on that.

P7L26. Figure 6 is misleading. I do not think this gives a good representation of the seasonal behavior, particularly if one wants to compare the three waves. Indeed, SW1 is clearly enhanced in winter (mainly no wave during the rest of the year), which is not felt in Fig. 6. Also, SW1 looks relatively stronger in Fig. 6 than in Fig. 4: in winter, relatively stronger than SW1; in May stronger than SW3; in late October, even SW3 dominates as seen in Fig 4, but not in Fig.6. Given the non-linear amplitude vertical grow, perhaps averaging amplitude relative seasonal anomalies at each altittude would work better.

P7L30. I do not agree that SW2 is a reasonable approximation of SW2, particularly above 92km. For example, according to Fig 4, SW1 + SW3 contribute around 30% during Jan-Feb. In early December, SW1 contributes more than 30%. Please, be more precise.

P7L31-32. Repeated.

P8L10. Is this also due to the non-linear SW2 - sPW1 interaction that excites SW1 preferentially (as compared to SW3) in the winter?

P8L26. It is true that CTMT's resolution smoothes the maxima and the minima but they can be inferred. However, the summer CTMT SW2 max is shifted one-two months in you measurements. Please, provide some explanation for this difference.

P8L27. Please, degrade the temporal and vertical resolution of your measurments to two months and 1,7km (as CTMT) and replace corresponding panels in Fig. 7. That way the comparison with CTMT would make more sense.

P8L30. different from -> before

P8L30-P90L1. I do not agree that the difference is due to an uneven sampling because that is not the case. Neither to the temporal resolution difference (that, on the other hand, should be seen once the radar temporal resolution is degraded) because that would just smear out the maximum instead of producing a temporal shift.

P9L2. Please, describe the major discrepancies for SW1.

P9L3-6. I do not really understand what new to Fig. 7 Figure 8 adds?

P9L19. neglected tidal components

P9L16. Discussing the overall yearly bias as compared to your amplitude estimations for SW1, SW2 and SW3 is misleading. It would be more useful to check the bias relative to the amplitudes for each month. For example, for SW3 and SW1 in August above 90km, the bias is 3-4 m/s, not bad, but the relative bias would be large or extremely large, respectively. In other words, estimated SW3 amplitude might be 50% biased and all estimated SW1 amplitude is not even SW1. Note that there is also the possiblility that CTMT is not fully correct.

P9L29. Please, comment also on possible leakage from waves of other periods on your estimated semidiurnal amplitudes.

P10L6. from five SMRs "located at roughly 3 longitudes"

P10L9. Contrary to "most"

P10L6. SW1 and SW3 do not enhance during SSWs, "as suggested by He et al. (2018a, b)".

P10L20. I find more useful to know when and how much SW2 is not a good approximation for the semidiurnal tide.

---

## Referee Comment (RC2) · Anonymous Referee #2 · 13 Mar 2019

This paper is innovative in the way that it uses longitudinally-distributed ground-based wind observations to get high temporal resolution and adequate spatial resolution to identify the sidebands (USB, LSB) of the Q16DW-SW2 interaction as distinct from M2, SW1, and SW3. The conclusion that previous space-based studies may have attributed USB and LSB to SW1 and SW2 and sometimes M2 is a very important and illuminating result.

In all, the paper is very well presented with new perspectives provided by the analysis and choice figures.

[Figure]

The interpretations in terms of polar vortex weakening and polar vortex classification during SSWs is also a very interesting and an important contribution. However, I wonder why correlations between USB, LSB, SW1 and SW3 with SPW and Q16DW are not reported, since the former set of waves is more directly/physically connected with SPW and Q16DW, rather than whether there is an SSW or not. It raises the questions: What is the connection between Q16DW and SSWs? For a split vortex, do S0 and SW4 replace SW1, SW3? Perhaps in the text you could explain why relationships with SPW and Q16DW are not reported, but SSW characteristics are used instead.

When comparing with CTMT, perhaps it would be beneficial to form 2-monthly means of the ground-based data so the comparison is more consistent?

line 23: consistency

---

## Author Comment (AC1) · 27 Mar 2019

Dear referee,

Thank you for your comments and suggestions. Enclosed please find our responses. The attachment zip archive comprises three pdf files, namely, the response to your comments, the revised manuscript, and a comparison between the revised manuscript and the previous version with the difference highlighted.

We hope our revisions and responses address your concerns properly. Any further

discussion and suggestion would, of course, be highly appreciated.

Respectfully, Maosheng He

Please also note the supplement to this comment:
https://www.atmos-chem-phys-discuss.net/acp-2018-1320/acp-2018-1320-AC1-supplement.zip

---

## Author Comment (AC2) · 27 Mar 2019

Dear referee,

Thank you very much for your valuable feedback. The manuscript was revised to introduce shortly, in Section 4 (before Section 4.1), the relationships among sPW, Q16DW, and SSWs, and to discuss the influence of the temporal resolution to the comparison with the empirical model. In the detached zip archive, please find the following three files, (1) the responses to your comments, (2) the revised manuscript, and (3) a

manuscript with revisions highlighted.

We hope our revisions and responses address your concerns properly.

Respectfully, Maosheng He

Please also note the supplement to this comment:
https://www.atmos-chem-phys-discuss.net/acp-2018-1320/acp-2018-1320-AC2-supplement.zip
* * *

---

## Author Response (AR1)

Dear Editor,

Thank you very much for processing our manuscript. Enclosed please find the detailed response to the comments, a revised manuscript, and a revised manuscript with the revisions highlighted. In the response, the comments from referee are in blue while our responses are in black. Our main revisions are:

(1)     According to the comments from both referees, we compare our results with CTMT the same resolution and discuss the influence of the resolution to our comparison at the end of section 5.2 shortly;

(2)     According to the comments from the first referee, the possibility of M2-sPW1 interaction is discussed at the end of section 4.2;

(3)     According to the comments from the second referee, the association between SSWs and 16-day waves are discussed shortly before section 4.1.

We hope our revisions and responses address the concerns of the referees properly.

Respectfully,

Maosheng He

**Response to the 1st referee's comments on "Mesospheric semidiurnal tides and near-12-hour waves through jointly analyzing five longitudinally-distributed specular meteor radars at boreal midlatitudes" by Maosheng He and Jorge L. Chau**

Maosheng He[1] and Jorge L. Chau[1]

[1]Leibniz-Institute of Atmospheric Physics at the Rostock University, Kühlungsborn, Germany

Dear referee,

Thank you for your valuable comments and suggestions. According to your comments, we made the following main revisions:

(1) after comparing our results with CTMT the same resolution, we discussed shortly the influence of the resolution to our comparison at the end of section 5.2;

(2) the possibility of M2-sPW1 interaction is discussed at the end of section 4.2.

Enclosed please find the detailed response to your comments, a revised manuscript, and a revised manuscript with the revisions highlighted. In the response, your comments are in blue while our responses are in black.

We hope our revisions and responses address your concerns properly. Any further discussion and suggestion would, of course, be highly appreciated.

Respectfully,

Maosheng He

[Figure]

**Figure 1.** Same plot as Figure 8 in the manuscript but using the SMR results after being averaged in a 2-month-wide sliding window

This manuscript reports on 6 semidiurnal and cuasi-semidiurnal tidal components, namely, SW2, SW1, SW3, M2 and the lower and upper side band interactions of SW2 and the 16-day PW (LSB and USB). These are derived from 5 radar measurements located at roughly three longitudes and 50N. The three 12hr components are further compared to results from CTMT, showing a relatively good agreement. They also study the tidal enhancement during SSWs. As in previous works, the authors

5   suggest that, due to their close period, aliasing between these waves may have lead to misinterpretation in semidiurnal tide measurements.

The paper is well written and easy to read, the methodology is generally adequate and the results are convincing. In general, the conclusions are not completely new and the results confirm previous works, but, in this one, a decomposition of the six waves is simultaneously performed using several radar measurements extending 7 years.

10   I think the manuscript deserves publication in ACP once the following comments are addressed.

*Main comments*

1. The title is misleading. It is true that 5 radars are used but they are not longitudinally distributed: only 3 longitudes are sampled. Please, change the title accordingly.

**Response:** In responding to this comment, we revise the title to specify that the radars are from three longitudinal sectors.

15   2. Radar measurements have a significantly better resolution than CTMT results ( 2months). Comparisons at similar temporal resolutions would make more sense. In that sense, the discussion in the text and panels a-f in Fig. 7 accordingly. On the other hand, that would additionally allow deleting Fig. 7 a-c panels, which are almost the same as j-l in Fig.4. That the phase does not change from year to year could just be mentioned in the text.

**Response:** The comparison at the same resolution is shown here in Figure 1 (cf., also the response to the other reviewer).

20  As expected, the correlation coefficients are slightly higher by just up to 0.01. In the revision, we introduce and discuss this

[Figure]

**Figure 2.** Same plot as Figure 7 in the manuscript but the SMR results are estimated at lower frequency resolution to contain the LSB and USB.

situation at the end of section 5.2. We hesitate to replace Figures 7 and 8 with their smeared version, because we intend to show how much different knowledge our result might bring.

Figures 4 and 7 displays <|a|> and |<a>|, respectively. Even though they might be somehow redundant, they facilitate the CA and comparison, individually. Further, the consistency between<|a|> and |<a>| indicates the year-to-year variation of $\hat{a}$ is negligible.

3. The authors report here (and also in previous works) that SW1 and SW3 are aliased with LSB and USB, respectively, in most previous tide studies from observations. I guess that CTMT is also that case. In the radar measurements with CTMT comparisons, why do you then compare SW1 and SW3 for both data-sets instead of SW1+LSB and SW3+USB for the measurements with SW1 and SW3, respectively, for the model?

**Response:** It is true that to compare the SMR tides with the CTMT tides+sidebands are not fair. In the revised section 5.2, filtering LSB and USB from SW1 and SW3 is discussed shortly in the comparison. Following this comment, we also produce

the same results as the Figure 7 from the manuscript but through wavelet at lower temporal resolution so that the resultant SW1 and SW3 contain the LSB and USB, respectively. The low resolution results are shown here in Figure 2.

However, we do not think it is fair to compare the SMR results here in Figure 2 with the CTMT because in our approach the sidebands (USB and LSB) contaminate the tides (SW3 and SW1) in a way different from that in CTMT. In CTMT with the static assumption, the contamination is basically the superposition of two waves, whereas in our approach and according to Equation (3) the sidebands might be both amplified or condensed. To replace the Figure 7 in the manuscript with Figure 2 here entails an evaluation of the amplification or condense. The evaluation entails further the prior knowledge of the sidebands. On the other hand, probably due to the difference in the contamination, the correlation coefficients between the SMR results and CTMT in Figures 2 (not shown) are not significantly higher than those in Figures 7.

Accordingly, the manuscript presents the tides after filtering the sidebands out to avoid the undetermined contamination from the sidebands.

4. In general, the manuscript sometimes misses the opportunity to explain the reasons for the tidal behavior. For example, there is no mention of the origin of the seasonal tidal variation.

**Response:** The main purpose of the current work is to develop a new approach based on a longitudinally-elongated network to decompose the near-12hr waves. With the decomposition, we focus on two particular topics, namely, the tidal activity during SSWs and the tidal seasonal variations. On the first focus, we illustrate and clarify the relations between SW1&SW3 and the secondary waves of the PW-SW2 interaction. On the seasonal variations, our main efforts are made for the comparisons with previous studies. The comparisons exhibit both consistency and difference. In section 5.1, we associate the consistent seasonal variations with a few references so that the readers are able to find the relevant discussions and the underlying reasons in more specific literature. The difference is illustrated in Sections 5.2 and 5.3, carefully and exhaustively. Although a part of the difference is explained at the end of Sections 5.1 and 5.2, the current manuscript has not presented complete physical explanations to the difference. We are still working on collecting convincing evidence for explaining the difference. Offering a convincing explanation is beyond the scope of the current work, but would be addressed in future works, independently.

Also, the reasons for the LSB and M2 dependence on the SSW classification could be explored further. Potential attribution to a planetary wave of particular wavenumbers might help.

**Response:** Yes, we share this point. In the revision, we add a short discussion of the possibility of the sPW1-M2 interaction following another of your comment.

A complete address of this relation is relying on understanding the relations between PWs and SSW. The relations have not yet been well understood so far, and we are working on investigating the relations with observations as well as models.

*Specific comments*

P2L1-2. MLT monitoring is not only possible with those two techniques. Please, expand.

P3L16. Define f and t

P4L1. are -> is

P3L5. 12hr -> 12hr period

P4L31.12.0h period

[Figure]

**Figure 3.** Same plot as Figure 8 in the manuscript but for the comparison between the tides at 40°N and those at 50°N according to CTMT.

P7L23. Please, specify that waves eventually dissipate.

P7L30. I do not agree that S2 is a reasonable approximation of SW2, particularly above 92km. For example, according to Fig 4, SW1 + SW3 contribute around 30% during Jan-Feb. In early December, SW1 contributes more than 30%. Please, be more precise.

P7L31-32. Repeated.

P8L30. different from -> before P9L19. neglected tidal components

P10L6. SW1 and SW3 do not enhance during SSWs, "as suggested by He et al. (2018a, b)".

P10L9. Contrary to "most"

**Response:** Thanks for above detailed suggestions. The manuscript is revised accordingly.

P2L4. Please, write single-point (in space or time).

**Response:** 'single-point' is explained with a sentence and a reference (Paschmann and Daly, 1998).

P28L8-9. this is not exact. Slowly precessing satellite measurements may have large temporal resolutions but still can distinguish temporal variations. Also, finer temporal resolution can be achieved in some particular cases (e.g. Li et al., JGR, doi:10.1002/2015JA021577, 2015).

**Response:** Li et al. (2015) entailed a assumption that the tides are static, namely, $\frac{\partial T}{\partial t}$=0 in the solar synchronous coordinate systems, in the one-month-wide smoothing window. To deal this ambiguity, we revise '*temporal variations*' to '*instantaneous temporal variations*', and specify the static assumption.

P3L2. Are the data available continuously from the years indicated?

**Response:** Yes, they are continuous with very rare short gaps. The data gaps are filled by interpolating. We are using liner interpolate, and our results are do not subject to the interpolating approach.

P3L28. Even if the correlation between 40 and 50N is high, what is the difference in amplitude of these modes in CTMT?

**Response:** Figure 3 here in the response displays the same plot as Figure 8 in the manuscript but for the comparison between the two latitude according to CTMT. The comparison can hardly be described qualitatively in details, and we think

the correlation coefficients are a good summary for the comparison. In the revision, we add one more reference for the readers interested in the meridian variation. Since there are works reporting the meridian variation, to quantify the meridian variation is not in our scope. We simply assume the difference is negligible for estimating the seasonal variations. Our assumption is different from the static assumption used in the analysis based on observations from single-spacecraft missions. To evaluate the assumptions comparatively is also a task in our plan. The evaluation entails an independent model with high resolutions in both time and space. We specified this situation in the revised Section 5.2.

P4L32. Shortly explain in the text why only those $m_k s$

**Response:** the reasons are that (1) at 11.6 and 12.4 hr, only these waves have ever been reported, (2) at 12.0hr, the three waves are climatically the strongest ones especially during SSWs. The whole Section serves as a detailed explanation. To link the explanation with our $m_k$ association, in the revision, we add on sentence at the beginning of the last paragraph on Section 2.2.

P5L2. Please, comment on possible aliasing with other period waves.

**Response:** the potential aliasing had been specified at the end of the previous paragraph as follows. In response to the current comment, we added specifically the term '*aliasing* ' in the discussion.

*...,in our wavelet analysis (cf., Grossmann et al., 1990), we set the* Morlet *factor to 128 so that the passed frequency band corresponds to 12.4±0.1hr, 11.6±0.1hr, and 12.0±0.1hr. These period bands are narrow enough to prevent power leakage* or aliasing *between each other.*

P6L28. Is it possible that the interaction between PW and M2 is the origin of LSB?

**Response:** good argument. In the revision, a paragraph is added in the end of Section 4.2 for a short discussion. To consolidate or exclude such possibility entails more observations, which is beyond the scope of the current work.

Although have never been proposed in existing literature, a secondary wave of the nonlinear interaction between sPW and M2 is an alternative explanation for the LSB signature at $T$=12.4hr and $m$=1, according to the resonance condition.

However, supporting the PW-SW2 interaction hypothesis are three types of evidence we have reported in case studies (e.g., He et al., 2017, 2018b):

(1) the triple occurrences of the three involved waves, the 16-day PW, SW2 and the LSB signature;

(2) the triple coherency among the involved waves;

(3) the USB associated with the PW-SW2 interaction.

However, none of the similar evidence has been reported in supporting the sPW-M2 interaction. On the contrary, in one case (He et al., 2018a), the LSB signature at $T$=12.4hr and $m$=1 was observed without an occurrence of significant M2.

Accordingly, we associate the LSB signature with the PW-SW2 interaction rather than the sPW-M2 interaction. Even though, the evidence from the case studies and our analysis are *NOT* sufficient to exclude the possibility of the existence of the sPW-M2 interaction.

P6 Why didn't you consider minor warmings? P7L5. There was a major final warming in March 2016 (Manney et al., 2016) but you find weakest M2 and LSB at that time. Please, explain why. Perhaps indicating your definition of non-SSW would help.

**Response:** In the revision, *non-SSW* is replaced by *non-major-SSW*. One sentence and three references are added for clarifying and specifying the definition.

According to the associated vortex breakdown, SSWs could be classified into three categories: major, minor and final SSWs.

The major SSW is associated with a complete disruption of the polar vortex which either splits into daughter vortices or displaces from the normal location. According to the behavior of the vortex, major SSWs could be further categorized into two types.

The minor SSW is less dramatic than the major. In the minor SSWs, the vortex weakens but does neither revers nor break down.

The final SSW occurs on the transition from prevailing westerly during winter to easterly during summer. Events in final SSWs might be associated with the dynamics of the seasonal transition instead of SSW. Therefore, we do not categorize the final SSWs into the types of vortex split and displacement. *The SSW in March 2016 is a final one*.

The PVW associated with minor SSWs is also presented. In the analysis shown in Figure 5, seven PVWs are clustered only into three groups: (1) major SSW with vortex split, (2) major SSW with vortex displacement, and (3) the rest. The rest inlcudes the minor and final SSWs.

P7L3. Please, provide your definition of PVW strength.

**Response:** it was defined by Zhang et al. (2014) as *'the minimum of westward zonal mean zonal wind at 70 and 48 km'*. Since this concept is not involved in the discussions, in the revision, a reference is added to specify this term.

P7L5. Is there any relationship between PW1 (associated to displacement events) and the strong LSB (m=1)?

**Response:** We assume the *PW1* is referring to the stationary planetary wave with zonal wavenumber 1 structure (sPW1). See the response to the comment above at 'P6L28'.

P7L18. I do not think it suggests to be more dominated by SSW but just that SSW have a significant effect.

**Response:** The world '*dominated*' is revised to '*characterized*'.

P7L24. temporal variations are similar except when the altitude of dissipation changes with the season. In general, it seems *that* does not apply to your waves (except SW3 in December, which apparently starts dissipating at lower altitudes than other years. Please, comment on that.

**Response:** We assume in the comment, '*that*' refers to '*temporal variations are similar*'. Accordingly, we revise the sentence as '*Such a simple vertical structure is associated with the fact that in Figure 3 the temporal variations, enhance and fading, of waves extent typically in a broad altitude range rather than at limited altitude levels.*'

We assume the last sentence is saying that the vertical gradient of SW3 is steeper than the others. If that is the case, we could hardly share this point. The vertical gradient of SW3 does not significantly greater than that of SW1. Careful quantitative comparison has to be done. Instead of going to the details, we release the decomposition results as a dataset so that the community could investigate this issue independently.

P7L26. Figure 6 is misleading. I do not think this gives a good representation of the seasonal behavior, particularly if one wants to compare the three waves. Indeed, SW1 is clearly enhanced in winter (mainly no wave during the rest of the year),

which is not felt in Fig. 6. Also, SW1 looks relatively stronger in Fig. 6 than in Fig. 4: in winter, relatively stronger than SW1; in May stronger than SW3; in late October, even SW3 dominates as seen in Fig 4, but not in Fig.6.

**Response:** the reviewer thought that there are discrepancies between Figures 4 and 6, e.g., SW1 looks stronger in Figure 6 than that in Figure 4. Therefore the reviewer comments that Figure 6 is not a good representation.

In fact, Figure 6 is consistent with Figure 4 if read the figures quantitatively referring to the limit of the y-axis in Figure 6a and the limit of the color range in Figure 4. The difference in the limit might have caused confusions. E.g., the y-axis in Figure 6a is shown in the range from 0 to 20m/s whereas the color for the SW1 in Figure 4d is from about 2m/s to 11m/s. As a result, the apparent annual variation in Figure 6a looks less visible than that in Figure 4d.

On the contrary, one purpose of Figure 6 is to eliminate the potential confusions by showing the amplitudes of the three components in the identical linear range.

Given the non-linear amplitude vertical grow, perhaps averaging amplitude relative seasonal anomalies at each altitude would work better.

**Response:** the average at separate altitudes is presented in Figures 4 and 7.

Here in Figure 6, we intend to present a one-dimensional description of the temporal variations for a comparison between different components. For our purpose, the variation at an arbitrary altitude works. We use the average because it is still more robust than that from an arbitrary altitude.

On the other hand, we also checked the seasonal variation at separate altitude, as well as averages in narrower altitude ranges, in plots similar to Figure 6, but did not find any interesting information more than the average in the whole altitude range.

The meaningfulness of the mathematical expectation or average is not immediately subject to the distribution of samplings. An example is the calculation of a national average income. Although the income distribution is rarely even or linear in most countries, the average income is still a meaningful value.

P8L10. Is this also due to the non-linear SW2 - sPW1 interaction that excites SW1 preferentially (as compared to SW3) in the winter?

**Response:** Our results do not support that the SW2-sPW1 interaction is responsible for the SW1 maximum in early December because the sPW1 maximizes climatically later than early December.

We do not think the interaction excites SW1 preferentially. In an early paper (He et al., 2017), we discussed and tried to explain the independent occurrence between LSB and USB. None of them has a permanent priority, and the SW3-like USB could also occur alone (He et al., 2017, 2018a).

P8L26. It is true that CTMT's resolution smooths the maxima and the minima but they can be inferred. However, the summer CTMT SW2 max is shifted one-two months in you measurements. Please, provide some explanation for this difference.

**Response:** We assume '*summer CTMT SW2 max*' refers to the maximum at September. If that is the case, there is not any shift between out SMR results and those from CTMT: the maximum occurs at September in both results of SMR and CTMT, in Figures 7b and 7h.

Even if there is a bias less than two months between the SMR and CTMT results, we would still say they are consistent since this bias is still below the temporal ambiguity.

[Figure]

**Figure 4.** A sketch of the influence of the temporal resolution to the location of the maximum.

P8L27. Please, degrade the temporal and vertical resolution of your measurements to two months and 1,7km (as CTMT) and replace corresponding panels in Fig. 7. That way the comparison with CTMT would make more sense.

**Response:** see the response to the 2nd main comment.

P8L30-P90L1. I do not agree that the difference is due to an uneven sampling because that is not the case. Neither to the temporal resolution difference (that, on the other hand, should be seen once the radar temporal resolution is degraded) because that would just smear out the maximum instead of producing a temporal shift.

**Response:** we regret to learn that the reviewer does not share our point. To convince the reviewer, we sketch in Figure 4 an example to illustrate the potential influence of the temporal resolution to the location of the maximum. In the figure, the blue line presents a time series at the resolution of half a month, while the red line presents the average of the blue line in every two months. The maximum occurs at 2.25 month on the blue line but 5 months on the red line. The difference is 2.75 month.

Of course, we do not have evidence to prove this is the real situation. Therefore, we are using a weak suggestive tone rather than a conclusive tone.

P9L2. Please, describe the major discrepancies for SW1.

**Response:** one sentence is added to describe the discrepancies. We hesitate to present more details because we are not able to explain the discrepancies in the current work.

P9L3-6. I do not really understand what new to Fig. 7 Figure 8 adds?

**Response:** Figure 8 uses the same information as used in Figure 7 but emphasizes a different perspective. Figure 8 has a very specific purpose for conveying the relationship between our result and the model. It quantifies intuitively their correlation.

P9L16. Discussing the overall yearly bias as compared to your amplitude estimations for SW1, SW2 and SW3 is misleading. It would be more useful to check the bias relative to the amplitudes for each month. For example, for SW3 and SW1 in August above 90km, the bias is 3-4 m/s, not bad, but the relative bias would be large or extremely large, respectively. In other words,

estimated SW3 amplitude might be 50% biased and all estimated SW1 amplitude is not even SW1. Note that there is also the possibility that CTMT is not fully correct.

**Response:** Although the first sentence of the current comment says that our discussion is *misleading*, after reading the whole comment, we believe the reviewer intends to say Figure 9 is *not readily* for comparing with the tidal amplitudes. Therefore, the reviewer suggests presenting the relative bias, namely, in percentage instead of the absolute bias in the unit of m/s.

We would say both presentations of the relative and absolute biases have their own advantage and disadvantages, like in any other analysis. While the reviewer listed only the advantage of the relative bias, there are following issues made us insist to use the absolute bias.

(1) We are not sure which tidal estimation shall we use as the divisor to calculate the percentage? Shall we use our estimation or CTMT result? From Figure 6, one could find that these two references are quite different.

(2) As the divisor, the tidal amplitude of all components in both references are often close to zero, which on the one hand amplifies the error propagation extremely and on the other hand yields the percentage is often extremely huge. These extremely huge values make it is technically challenging to display all values in a figure. On the contrary, both the absolute value and the error propagation are robust in the absolute bias.

P9L29. Please, comment also on possible leakage from waves of other periods on your estimated semidiurnal amplitudes.

**Response:** this issue has been addressed early in Section 2.2 when the *Morlet* wavelet factor was specified as 128. The wavelet analysis works as band-pass filtering. The leakage is restricted in $\Delta T = \pm 0.1$hr, namely, $11.6 \pm 0.1$hr, $12.0 \pm 0.1$hr, and $12.4 \pm 0.1$hr, respectively.

P10L6. from five SMRs "located at roughly 3 longitudes"

**Response:** this information is specified in the first sentence of the revised conclusion.

P10L20. I find more useful to know when and how much SW2 is not a good approximation for the semidiurnal tide.

**Response:** The last sentence is revised to be more objective and specific. The criterion for *a good approximation* might be subjective and flexible.

We hope our revisions and responses address your concerns properly.

Respectfully,

10  Maosheng He

This paper is innovative in the way that it uses longitudinally-distributed ground-based wind observations to get high temporal resolution and adequate spatial resolution to identify the sidebands (USB, LSB) of the Q16DW-SW2 interaction as distinct from M2, SW1, and SW3. The conclusion that previous space-based studies may have attributed USB and LSB to SW1 and SW2 and sometimes M2 is a very important and illuminating result.

5    In all, the paper is very well presented with new perspectives provided by the analysis and choice figures.

The interpretations in terms of polar vortex weakening and polar vortex classification during SSWs is also a very interesting and an important contribution. However, I wonder why correlations between USB, LSB, SW1 and SW3 with SPW and Q16DW are not reported, since the former set of waves is more directly/physically connected with SPW and Q16DW, rather than whether there is an SSW or not. It raises the questions: What is the connection between Q16DW and SSWs? Perhaps in the

10   text you could explain why relationships with sPW and Q16DW are not reported, but SSW characteristics are used instead.

**Response:** a paragraph is added to Section 4, immediately before the title of Section 4.1, to discuss the triple association among Q16DW, SW2 and the secondary wave and the association between Q16DW and SSWs.

We have not discussed sPW and Q16DW in details mainly because the current work is observation-orientated. The current manuscript is organized largely as following,

15   (1) propose a new approach;

(2) describe the results, either consistent or inconsistent with previous results;

(3) discuss the potential explanations.

In describing the results, we note that the 11.6hr and 12.4hr oscillations often enhanced after SSWs (cf., Figure 3 in the manuscript). Therefore, to investigate the association with SSWs is intuitive and straightforward for us. Besides, there are two

20   other reasons that motivate us to explore the SSW association. The first is that during SSWs reported were enhancements of five near-12hr waves, namely, SW1, SW3, M2, the LSB, and USB. The other is that SSWs provide a good epoch reference.

Physically, the association of the parent waves and secondary waves should definitely be explored in a future effort.

In the added paragraph, we refer some studies reporting the triple association among Q16DW, SW2 and the secondary wave and the association between Q16DW and SSWs. These works used single radar approaches. We are also preparing an

25   independent manuscript investigating the association between PWs and SSW through multi-radar approaches.

For a split vortex, do S0 and SW4 replace SW1, SW3?

**Response:** After vortex splitting, sPW-2 amplifies. If sPW interacts with SW2, S0 and SW4 would be generated according to the resonance conditions. However, instead of supporting the sPW-SW2 interaction, the current manuscript reports evidence supporting the traveling Q16DW-SW2 interaction. Namely, the potential secondary waves of sPW1-SW2 do not enhance

30   during SSWs, instead, those of Q16DW-SW2 interaction does.

When comparing with CTMT, perhaps it would be beneficial to form 2-monthly means of the ground-based data so the comparison is more consistent?

**Response:** we share the point that to compare results at the same resolution is fairer than what we showed. Here, in Figure 1, we display the suggested comparison. The correlation coefficients of all components are higher than those displayed in the

[Figure]

**Figure 1.** Same plot as Figure 8 in the manuscript but using the SMR results after being averaged in a 2-month-wide sliding window

original Figure 8, by up to 0.01. This situation is discussed at the end of revised Section 5.2. We hesitate to replace Figures 7 and 8 with their smeared version because our main purpose here is to emphasize the difference rather than the consistency.

*Specific comments*

Page6,line 23: consistency

5    **Response:** revised.

[revised manuscript text omitted]